# Implications of Transition towards Manufacturing on the Environment: Saudi Arabia's Vision 2030 Context

**Nasreen Alfantookh, Yousif Osman * and Isam Ellaythey**

Department of Economics, College of Business, King Faisal University, Al-Ahsa 31982, Saudi Arabia
* Correspondence: yyoysif@kfu.edu.sa

**Abstract:** This study is based on the idea that Saudi Arabia's Vision 2030 considered the achievement of economic diversification is very crucial for the economy. In turn, this target requires a sustained increase in the contribution of the manufacturing sector in Gross Domestic Product (GDP). At the same time, the transition towards industrialization might trigger high rates of $CO_2$ emissions, due to the escalated manufacturing demand for primary energy consumption (specifically fossil fuel). Ultimately, the high rates of $CO_2$ emissions would have severe environmental consequences, such as environmental degradation. These environmental consequences might be more dangerous in a country extensively dependent on oil, such as Saudi Arabia. The study aims to investigate the manufacturing and environment nexus in an attempt to explore the validity of the inverted U-shaped curve, the so-called Kuznets hypothesis, during 1971–2021. Applying the econometric model autoregressive distributed lag (ARDL), the findings of the study do not show evidence supporting the validity of an inverted U-shaped Kuznets function in Saudi Arabia during the period of the study. Furthermore, the short-term results do not confirm the impact of increasing manufacturing on $CO_2$ emissions. However, there are indications of positive effects, although limited, in the long-term.

**Keywords:** manufacturing; environmental Kuznets curve; ARDL; Saudi Arabia's Vision 2030

## 1. Introduction

Around three-quarters of global greenhouse gas emissions come from the burning of fossil fuels for energy. Despite producing more and more energy from renewables each year, the global energy mix is still dominated by coal, oil, and gas. Not only does most of our energy—84% of it—come from fossil fuels, but we also continue to burn more each year: total production has increased from 116,214 to 136,761 TWh in the last 10 years (Roser and Ritchie 2020). The types of pollution resulting from oil production are various, including water, soil, and air, and these effects are usually long-lasting. There are many reasons to believe that oil production is responsible for greenhouse gas (GHG) emissions. $CO_2$ emissions count for a substantial proportion of GHGs and are responsible for increasing global warming (United States Environmental Protection Agency 1990–2019; Mahmood et al. 2020). Economic growth is a basic need for any economy, but its environmental effects should not be ignored. The results of the study of data for the period 1968–2014 in investigating the environmental effects of economic growth and energy consumption in Saudi Arabia indicate that economic growth and energy consumption contribute to carbon dioxide emissions in the long- and short-term. This means that the Kingdom's increased economic growth has a social cost to the economy in terms of pollution emissions (Mahmood 2019). Saudi Arabia needs the oil price to be at 100 USD a barrel to achieve financial balance, and hence, the current low prices necessarily push for an initiation in an economic diversification policy, and it is recommended to use oil revenues for investing in various economic sectors, especially manufacturing, to boost added value to the national economy (Bokhari 2017).

Despite previous studies, Saudi Arabia is committed to contributing to the Paris Agreement in making the world an environmentally friendly place. In the Vision 2030, the

Kingdom targets low air, sound, water, and soil pollution in the domains of the strategic objectives of national industrial development and national transformation programme, designed to offer a fulfilling and healthy life (Saudi Vision 2030 2019; Mahmood et al. 2020). Avoiding the curse of natural resources and maintaining a sustainable environment and economic growth for a country that is one of the largest oil-exporting countries is a difficult challenge, where the most difficult is the contribution of the manufacturing sector as one of the sources of income and increase in GDP, as industrialization means the use of more natural resources.

### 1.1. Problem Statement

The economy is thriving when there is an increase in manufacturing industries. At the same time, manufacturing contributes to various types of environmental pollutants, including carbon dioxide emissions, due to high rates of energy consumption, specifically fossil fuels. Knowing the fact that Saudi Arabia adopted an initiative towards manufacturing-oriented diversification in its 2030 vision, this fact is at odds with the objectives of the United Nations Framework Convention on Climate Change and the Paris Climate Agreement. Is the EKC hypothesis applicable to Saudi Arabia?

The trend towards manufacturing-oriented economic growth does not come without damage to the environment. Therefore, it is very important to find out to which extent the diversification of the Saudi economy through expansion of manufacturing affects the environment.

### 1.2. Research Question

In light of the contradicted relation between increased contributions of manufacturing industries in the economy and the safety of the environment, the study tried to find answers to the following questions:

1.  Is the EKC hypothesis valid for the case of Saudi Arabia?
2.  How far does the manufacturing-oriented economic growth affect the long-term environmental pollution?

### 1.3. Research Aim and Objective

The study aims to:

1.  Show the nexus of manufacturing-oriented diversification and the environment in Saudi Arabia.
2.  Investigate the validity of the inverted U-shaped Kuznets hypothesis in the Saudi economy.

### 1.4. Research Hypotheses

This research addressing two hypotheses:

**Hypothesis 1 (H1).** *Manufacturing-oriented growth in the Saudi economy does not result in an increase in long-term environmental pollution.*

**Hypothesis 1 (H2).** *The inverted U-shaped Kuznets hypothesis is applicable to the state of the Saudi economy.*

## 2. Literature Review

Historically, economic models about growth and diversification of the economy have been associated with the sustained increase of the manufacturing sector's contribution to the GDP. This fact is at odds with most claims of environmental economists and environmental activists, who tend to link the manufacturing-induced growth, and hence, the economic diversification with high rates of carbon dioxide emissions, which eventually lead to resource depletion and environmental degradation.

One of the most prominent models of the economic theory (by Simon Kuznets 1955) hypothesized the relationship between various indicators of environmental degradation and per capita income. According to Kuznets, " . . . .in the early stages of economic growth pollution emissions increase and environmental quality declines, but beyond some level of per capita income the trend reverses. This implies that environmental impacts or emissions per capita are an inverted U-shaped function of per capita income" (Stern 2018); the so-called environmental Kuznets curve EKC. See Panels (a) and (b) in Figure 1.

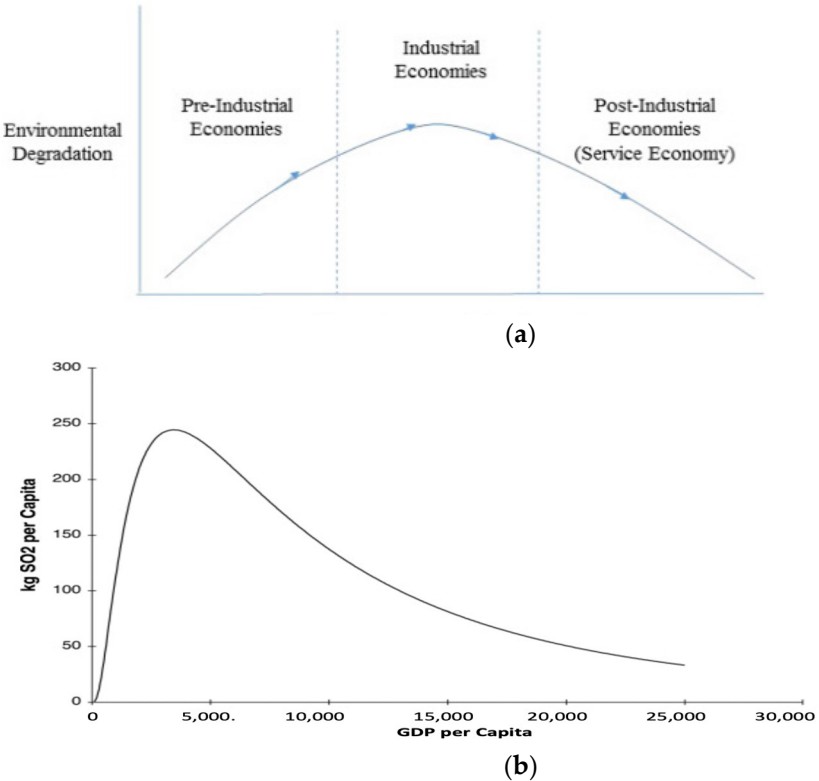

(**a**)

(**b**)

**Figure 1.** U-shaped EKC. Panel (**a**): Stages of economic growth. Panel (**b**): Emissions per capita and GNP per capita. Source: Stern (2018).

The validity of the EKC hypothesis has already been empirically tested by a wide spectrum of works, even for the case of Saudi Arabia. Therefore, the following paragraphs present a literature review about the validity of the inverted U-shaped function hypothesis (EKC).

Holtz-Eakin and Selden (1995) used panel data for 130 countries for the period 1951–1986 to estimate the relationship between carbon dioxide emissions and GDP. They concluded that economic development will not significantly change the annual or cumulative flow of carbon dioxide emissions in the future and the marginal propensity to emit $CO_2$ diminishes as economies develop. The validity of the EKC hypothesis has been strongly supported by findings of a study about a sample of 10 MENA countries for the period 1990–2010 (Fakhri et al. 2015).

Shahbaz et al. (2012) tried to investigate the existence of the EKC at an individual country level. Taking the case of Pakistan during the period 1971–2009, they came up with the finding that more energy consumption increases carbon dioxide emissions in the short and long run as well. While the openness to trade reduces carbon dioxide emissions in the long run, it is insignificant in the short run. Additionally, a similar contribution tried to investigate the existence of the EKC in Tunisia for the period 1971–2010. The results of this study support the validity of the EKC hypothesis (Shahbaz et al. 2014).

Furthermore, the validity of the EKC hypothesis was confirmed by different empirical contributions. For example, Al-Mulali et al. (2016), in their study about Kenya for the

period 1980–2012; Sinha and Shahbaz (2018), in their study about India for the period of 1971–2015; Ali et al. (2021), in their study about Malaysia for the period 1971–2012; and Pata (2018) in his study about Turkey for the period 1974–2013.

In contrast, other empirical contributions do not confirm the validity of the EKC hypothesis. For example, Al-Mulali et al. (2016), in their study about Kenya for the period 1980–2012; Solarin and Lean (2016), in their study about China and India for the period 1965–2013; Twerefou et al. (2016), in their study about Ghana for the period 1970–2010; Solarin and Lean (2016), in their study about China and India for the period 1965–2013; and Özden and Beşe (2021), in their study about Australia for the period 1960–2014.

In the context of Saudi Arabia, the literature has shown different contributions investigating the validity of the EKC hypothesis. For instance, Alshehry and Belloumi (2015) conducted a study about the causal relationship between energy consumption, energy prices, carbon dioxide emissions and economic growth during the period 1971–2010. The results of the study showed that the share of energy consumption in explaining economic growth is minimal in Saudi Arabia. Alkhathlan and Javid (2015) conducted a study about the impact of total oil consumption as well as oil consumption in the transport sector on the environmental quality of Saudi Arabia over the period of 1971 to 2013. The results of this study revealed that the trend is non-linear and stochastic for carbon emissions in both models; the elasticity of carbon emissions with respect to total oil consumption and transport oil consumption is positive and significant.

Another contribution investigated the validity of EKC hypothesis in Saudi Arabia for the period 1970–2014 by considering three channels: volume of production, sector value added to GDP and technological innovation. This study found that GDP growth and $CO_2$ emissions are positively and linearly associated in both the short and long run. This finding strongly opposes the hypothesis of the EKC (Samargandi 2017).

Recently, Mahmood et al. (2020) tried to estimate the asymmetrical impacts of the oil sector on $CO_2$ emissions. They came out with the result that the GDP per capita has a positive and inelastic effect on carbon dioxide emissions. The inelastic effect shows that $CO_2$ emissions per capita are increasing less than proportionally with a proportionate increase in income, but still, increasing income is responsible for environmental degradation by releasing more $CO_2$ emissions. In addition, Mahmood et al. (2020) investigated the energy and growth cubic relationship in Saudi Arabia in the long-term to verify the N-shaped energy EKC for the period 1970–2019. The findings of this study confirmed the validity of the energy–EKC hypothesis in the short, as well as in the long term. Most recent work by AlKhars et al. (2022) have been literature reviews of the EKC in the six GCC countries for the period between 2010 and 2020. Surprisingly, the findings of this study revealed mixed results about the support of the EKC hypothesis in the GCC area.

Most recently, Hamieh et al. (2022) published an article about the quantification and analysis of the $CO_2$ footprint from industrial facilities in Saudi Arabia. This study aims to provide a vital resource for researchers and policymakers who seek to reduce greenhouse gas emissions by promoting renewable energy, improving the efficiency of existing fossil-fuel-based industries, and evaluating the potential of carbon capture utilization and storage (CCUS) in KSA. The study followed the 2006 IPCC guidelines to estimate the $CO_2$ emissions from different sectors, which were mostly based on default emission factors. The main finding of the study stated that stationary industrial sources in the Kingdom of Saudi Arabia are responsible for more than 70% of the country's total emissions.

It is necessary to mention that the vast majority of the above-reviewed empirical contributions used the autoregressive distributed lag (ARDL) model, along with differences in the model specifications, time span, explanatory variables, sample scope, and shapes of the EKC. As far as the EKC studies in Saudi Arabia or related regions have shown an association between emissions and growth, this study will attempt to investigate the evidence of the manufacturing–EKC hypothesis in Saudi Arabia in a different context; that is, the regression of environmental deterioration with the manufacturing value added only, including the corresponding quadratic (ad cubic) terms. As far as the reviewed literature

does not cover the manufacturing–environment nexus per se in Saudi Arabia, this could be the contribution of this study.

## 3. Research Methodology

### 3.1. Data

The sample period covers the years from 1971 to 2021 of the Saudi economy, where secondary data published by the World Bank were used. Our variables are shown in Table 1.

**Table 1.** Variables' definition and description.

| Variable | Notation | Description | Unit |
|---|---|---|---|
| $CO_2$ emissions | $LCO_2$ | Carbon dioxide emissions include carbon dioxide produced during the consumption of solid, liquid, and gas fuels and gas-flaring | Metric tons per capita |
| GDP per capita | LGDP | GDP per capita equal to Real GDP/Population | Constant 2021 US$ |
| Energy use | LNG | Using the energy before transformation to other end-use fuels | Kg of oil equivalent per capita |
| Manufacturing value added per capita | LMV | Is calculated by dividing LMV by population | Constant 2021 US$ |
| Trade | LTR | Equals to all exports and imports of goods and services measured as a share of GDP | % of GDP |
| Foreign direct investment net inflows | LFDI | FDI net inflows are the value of inward direct investment made by non-resident investors | % of GDP |

Source: World Bank.

Table 2, present the Statistical description of variables, we entered the logarithm into (LCO2, LGDP, LMV and LNG) in order to gauge the homogeneity of the values of these variables, while the rest of the variables were basically calculated by percentage or per capita (Gujarati 2009). When we applied logs, the distribution was better behaved. Figure 2, show the graph of the underlying variables.

**Table 2.** Statistical description of variables.

| | $LCO_2$ | LGDP | LMV | LTR | LFDI | LNG |
|---|---|---|---|---|---|---|
| Mean | 11.382 | 12.239 | 25.047 | 74.725 | 1.1082 | 4.603 |
| Median | 11.1401 | 8.371 | 20.320 | 66.343 | 0.4961 | 7.151 |
| Maximum | 14.878 | 8.970 | 21.346 | 102.67 | 7.2321 | 7.524 |
| Minimum | 8.3449 | 8.1922 | 19.292 | 47.742 | −6.995 | 5.863 |
| Std. Dev. | 1.6714 | 0.2426 | 0.5792 | 11.012 | 2.6321 | 0.475 |
| Skewness | 0.4155 | 0.8563 | −0.0608 | 0.5558 | 0.0524 | 1.041 |
| Kurtosis | 2.0868 | 2.1114 | 1.7571 | 3.2636 | 3.5848 | 2.783 |
| Jarque-Bera | 1.9571 | 6.738 | 1.3981 | 3.7478 | 2.3140 | 9.463 |
| Probability | 0.2696 | 0.0162 | 0.3744 | 0.0943 | 0.2186 | 0.0032 |
| Sum | 500.30 | 373.12 | 894.62 | 2925.2 | 44.322 | 307.3 |
| Sum Sq. Dev. | 141.126 | 2.9737 | 16.947 | 6128.2 | 349.98 | 11.43 |
| Observations | 51 | 51 | 51 | 51 | 51 | 51 |

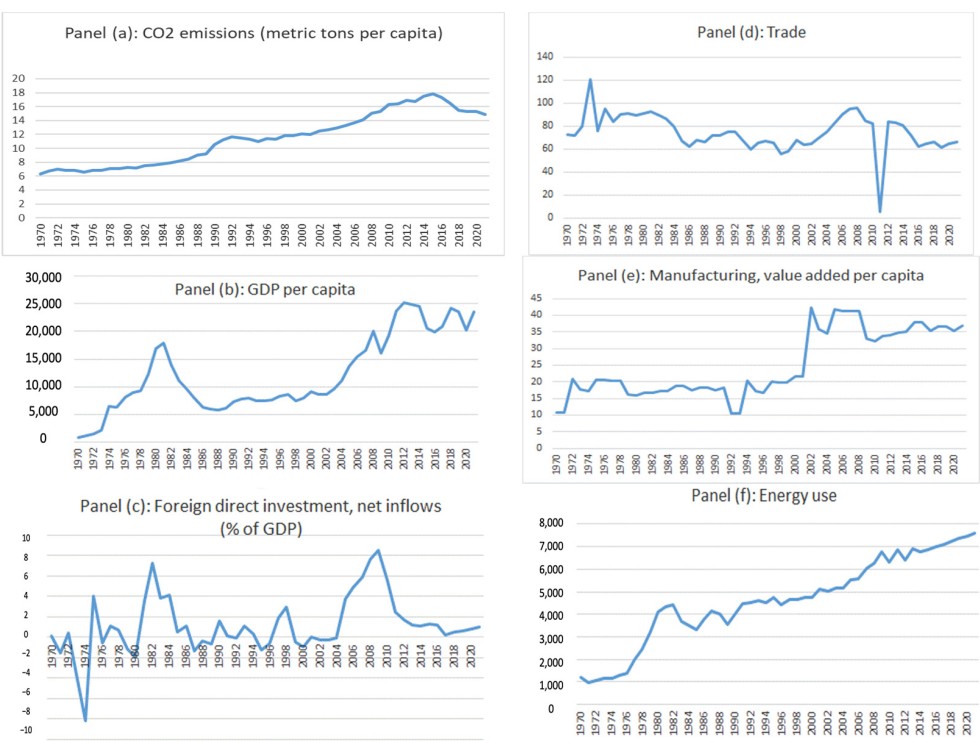

**Figure 2.** Underlying variables.

### 3.2. Model of Study

The methodology of econometrics to test the hypotheses of the study by applying the following ARDL model was used in this study, one of the most recent dynamic approaches that takes the element of time into account with the aim of identifying the short- and long-term relationships among the variables, and to provide evidence in favor of the Environmental Kuznets Curve (EKC) hypothesis in Saudi Arabia during the period of study.

Equation (1) explains how the linkage between $CO_2$ emissions and energy consumption (use), economic growth, manufacturing, trade, and FDI can be specified, as follows:

$$LCO_2 = f\,(LGDP, LGDP^2, LMV, LTR, LFDI, LNG) \tag{1}$$

### 3.3. Unit-Root Test

To test the stationarity of time-series (Pesaran 2015) for the study variables, we used the unit root test of Phillips and Perron (1988) (PP test), and the Augmented Dickey and Fuller (Dickey and Fuller 1979) (ADF test). Regarding the results shown in Table 3, our variables are a mix of integrated of different orders, that is, order zero, I(0) and order one, I(1), meaning that we were able to use the ARDL model to identify the short- and long-run relationships between the study variables (Nkoro and Uko 2016).

**Table 3.** PP test and ADF test results.

| | PP Test | | | | ADF Test | | | |
|---|---|---|---|---|---|---|---|---|
| **Series** | **Level** | | **First Difference** | | **Level** | | **First Difference** | |
| | *t*-Statis. | Prob. | *t*-Statis. | Prob. | *t*-Statis. | Prob. | *t*-Statis. | Prob. |
| $LCO_2$ | −2.1300 | 0.2344 | −6.3804 | 0.0000 *** | −2.1300 | 0.2344 | −6.3804 | 0.0000 *** |
| LNG | −1.6965 | 0.4258 | −5.0449 | 0.0002 *** | −1.6965 | 0.4258 | −5.0449 | 0.0002 *** |
| LFDI | −3.5190 | 0.0121 ** | −9.9726 | 0.0000 *** | −3.5190 | 0.0121 ** | −9.9726 | 0.0000 *** |
| LGDP | −1.4204 | 0.5635 | −5.0627 | 0.0001 *** | −1.4204 | 0.5635 | −5.0627 | 0.0001 *** |
| $LGDP^2$ | −0.2607 | 0.0924 | −3.1733 | 0.0021 *** | −0.3548 | 0.7589 | −3.2659 | 0.0011 *** |
| LMV | −0.4083 | 0.8986 | −7.1287 | 0.0000 *** | −0.4083 | 0.8986 | −7.1287 | 0.0000 *** |
| LTR | −3.0570 | 0.0376 ** | −10.5649 | 0.0000 *** | −3.0570 | 0.0376 ** | −10.5649 | 0.0000 *** |

Notes: *** and ** denote significance at the 10%, and 5% levels, respectively.

### 3.4. Econometric Model

Equation (1) can be re-written, explicitly, as follows:

$$LCO_{2t} = \beta_0 + \beta_1 LGDP_t + \beta_2 (LGDP_t)^2 + \beta_3 LMV_t + \beta_4 LTR_t + \beta_5 LFDI_t + \beta_6 LNG_t + \varepsilon \quad (2)$$

According to the EKC model, it is expected that economic activity is stimulated with a positive relationship between energy consumption and carbon dioxide emissions. Thus, theoretically, it is expected that $\beta_6 LNG_t > 0$. The EKC hypothesis reveals that $\beta_1 LGDP_t > 0$ while the sign of $\beta_2 (LGDP_t)^2$ should be negative ($\beta_2 (LGDP_t)^2 < 0$). Besides, the linkage between manufacturing, trade, FDI, and carbon dioxide emissions is positive, so it is expected that both $+ \beta_3 LMV_t$, $\beta_5 LFDI_t$, and $\beta_4 LTR_t > 0$.

One of the most important problems facing standard models and regression analysis is the linear correlation (multi-collinearity) between the independent variables. That means the failure to fulfil one of the assumptions of the OLS method. In the regression model, there are correlation linkages between the independent and dependent variables, which makes it tricky to disconnect their individual effects (Zahari et al. 2014). There are some indicators to detect this problem (Wooldridge 2015). Among these indicators, we can use the correlation matrix between the independent variables and the variance-inflating factor (VIF). A VIF greater than 10 and a pair-wise or zero-order correlation coefficient between two regressions more than 0.8 reveals multi-collinearity (Gujarati 2009).

The Correlation Matrix indicates that there are no multi-collinearity linkage independent variables as long as the Pearson correlation coefficient is less than (0.8), and the results in Table 4 show that between LNG and LGDP, LGDP$^2$ has been dropped.

**Table 4.** Correlation matrix results—pair-wise correlations.

|  | LCO2 | LGDP | LGDP$^2$ | LMV | LTR | LFDI | LNG |
|---|---|---|---|---|---|---|---|
| LCO$_2$ | 1.00000 |  |  |  |  |  |  |
| LGDP | 0.2656 | 1.00000 |  |  |  |  |  |
| LMV | 0.2988 | −0.6676 | 1.00000 |  |  |  |  |
| LTR | 0.49265 | 0.6231 | −0.1960 | 1.0000 |  |  |  |
| LFDI | 0.1913 | −0.2462 | 0.4239 | 0.2516 | 1.0000 |  |  |
| LGDP$^2$ | 0.2653 | 0.9999 | −0.6699 | 0.6209 | −0.2505 | 1.0000 |  |
| LNG | 0.2739 | −0.7324 | 0.8966 | −0.2918 | 0.4669 | −0.7344 | 1.0000 |

To set the optimal lag longitude needed for the model's variables, we used different criteria, like (AIC), (HIC) and (SIC). Table 5 shows lag from 1 to 3 for which the values of most of these lag length criteria are minimized by asterisks. Schwarz's criterion shows that the lag period for the lowest value is one period, while the Akaike's criterion shows that the lag period is three periods.

**Table 5.** Criterion for model selection.

| lag | AIC | SC | HQ |
|---|---|---|---|
| 0 | 16.55991 | 16.76889 | 16.63601 |
| **1** | 7.890607 | 9.144441 * | 8.347184 * |
| **2** | 7.853450 | 10.15214 | 8.690507 |
| **3** | 7.848290 * | 11.19185 | 9.065829 |

* indicates lag order choices by the standard. Akaike information criterion (AIC). Schwarz Criterion (SC). Hannan-Quinn (HQ).

We are conducted Autoregressive Distributed-Lag (ARDL) in this research. The ARDL model is one of the most popular recent dynamic approaches that takes the element of time into account. With the aim of identifying short- and long-run rapport among the variables, as well as the speed of the system's convergence to equilibrium, we analysed the long-run rapport between variables based on time-series data. This model consists

of two components: (1) Autoregressive (AR), that is, a model depending on its lagged values, meaning it uses the dependent variable as a lagged independent variable; and (2) Distributed Lagged (DL), indicating that the dependent variable is influenced also by the changes in the independent variables and their lagged values.

The following equation shows the ARDL model for our study, as follows:

$$
\begin{aligned}
dLCO2_{2t} = {} & \alpha_0 + \alpha_1\, LCO2_{t-1} + \alpha_2\, LGDP_{t-1} + \alpha_3\, LGDP^2{}_{t-1} + \alpha_4\, LMV_{t-1} + \alpha_5\, LTR_{t-1} + \alpha_6\, LFDI_{t-1} \\
& + \sum_{j=1}^{p}\beta_{1j}\Delta LCO2_{t-j} + \sum_{j=0}^{q}\beta_{2j}\Delta LGDP_{t-j} + \sum_{j=0}^{n}\beta_{3j}\Delta LGDP^2{}_{t-j} + \sum_{j=0}^{r}\beta_{4j}\Delta l_{LMV_{t-j}} \\
& + \sum_{j=0}^{m}\beta_{5j}\Delta LTR_{t-j} + \sum_{j=0}^{k}\beta_{6j}\Delta LFDI_{t-j} + \mu_{it}
\end{aligned}
\tag{3}
$$

where (d) refers to the first-difference operator; $p$, $q$, $r$, $m$, $n$ and $k$ indicate lags; ($\alpha_1$–$\alpha_6$) refers to long-run parameters; ($\beta_1$–$\beta_6$) refers to short-run parameters; ($\alpha_0$) refers to the intercept; and ($\mu_t$) refers to the error term.

The short-run effects were estimated from the following:

$$
\begin{aligned}
dLCO2_{2t} = {} & \alpha_0 + \sum_{j=1}^{p}\beta_{1j}\Delta LCO2_{t-j} + \sum_{j=0}^{q}\beta_{2j}\Delta LGDP_{t-j} + \sum_{j=0}^{n}\beta_{3j}\Delta LGDP^2{}_{t-j} + \sum_{j=0}^{r}\beta_{4j}\Delta l_{LMVt-j} + \\
& \sum_{j=0}^{m}\beta_{5j}\Delta LTR_{t-j} + \sum_{j=0}^{k}\beta_{6j}\Delta LFDI_{t-j} + \mu_{it}
\end{aligned}
\tag{4}
$$

$\varphi ECT_{t-1}$ represents the speed of adjustments towards the long-run equilibrium, meaning that if the system is moving out of equilibrium in one direction, then it will pull it back to equilibrium (Ali et al. 2021). "A positive coefficient indicates a divergence, while a negative coefficient indicates convergence. If the estimate of ECt = 1, then 100% of the adjustment takes place within the period, or the adjustment is instantaneous and full, if the estimate of ECt = 0.5, then 50% of the adjustment takes place each period/year. ECt = 0, shows that there is no adjustment, and to claim that there is a long-run relationship does not make sense anymore" (Nkoro and Uko 2016).

Selecting a model of ARDL by using the (SIC) criterion to determine the lags did not give a significant coefficients estimation result for the (LMV) variable in the short run and also did not give a significant coefficients estimation result for all independent variables in the long run.

### 3.5. Short-Run Relationship Estimation Results

Estimates in Table 6, indicate in the short run show in Table 6, that (LGDP) significantly affects carbon dioxide emissions in Saudi Arabia, and the parameter (LGDP) is a positive, which means that there is a direct linkage between GDP and emissions. Whenever the gross domestic product increases by one unit, carbon dioxide emissions increase by approximately (6%), and this is attributed to other economic activities that cause those emissions. As for the variable (LFDI), it has a statistically significant estimate of emissions, with a negative sign parameter. Whenever (LFDI) decreases by one unit, (LCO2) increases by approximately (0.19%), and this may be due to some restrictions that apply and some environmental policies to limit the increase in emissions and preserving the environment. As for the two variables (LMV) and (LTR), the estimates show that they reign no effect on carbon dioxide emissions in the short run.

**Table 6.** ARDL model estimation in the short-run and long-run.

| Short Run Estimates | | | | |
|---|---|---|---|---|
| **Variable** | **Coefficient** | **Std. Error** | ***t*-Statistic** | **Prob.** |
| $LCO_{2(-1)}$ | 0.772469 | 0.101941 | 7.577601 | 0.0000 |
| LGDP | 5.835308 | 1.880034 | 3.103831 | 0.0038 |
| $LGDP_{(-1)}$ | −4.034057 | 1.979621 | −2.037793 | 0.0492 |
| $LGDP^2$ | 0.595700 | 0.383425 | 1.553628 | 0.1293 |
| LMV | 0.013262 | 0.016283 | 0.814464 | 0.4209 |
| LTR | −0.141843 | 0.061535 | −2.305060 | 0.0272 |
| LFDI | 0.185506 | 0.053963 | 3.437639 | 0.0015 |
| C | −0.264716 | 0.057598 | −4.595885 | 0.0001 |
| **Long-Run Estimates** | | | | |
| LGDP | 7.916522 | 4.317275 | 1.833685 | 0.0752 |
| $LGDP^2$ | 2.618109 | 1.399359 | 1.870934 | 0.0697 |
| LMV | 0.058286 | 0.071057 | 0.820265 | 0.4176 |
| LTR | 0.191902 | 0.328451 | 0.584263 | 0.5628 |
| LFDI | 0.216511 | 0.617325 | 0.235689 | 0.0953 |
| C | −132.0875 | 65.80327 | −2.007308 | 0.0525 |

With regard to the results of the error correction model (ECM), the error correction term ($ECT_{t-1}$) is highly significant at the specified level of significance, 5%, with the expected negative sign, and this indicates the existence of a short-term equilibrium relationship cointegration relationship among the model variables. The coefficient of (*ECT*) approximately equals to (0.26). This means that deviations in the short-run are corrected by approximately 26% within one year towards the short-run equilibrium relationship.

### 3.6. Long-Run Relationship Estimation Results

Estimates of long-term results in Table 6, reveal that there is a statistically significant and non-negative economic link for the (LGDP), ($LGDP^2$), LTR and (LFDI) with ($LCO_2$), and this means that the relationship is positive in the long-term. Whenever ($LGDP^2$) increases by one unit, ($LCO_2$) increases approximately by (2.6%), so the convex function becomes ($7.917 + 2.618LGDP^2$), (LTR) increases by one unit, ($LCO_2$) increases by approximately (0.2%); and whenever (LFDI) increases by one unit, ($LCO_2$) increases by approximately (0.22%). These results demonstrate the opposite of the expectations of the study hypotheses, in that industrialization will not affect environmental pollution in the long term, and that the hypothesis of the environmental Kuznets curve corresponds to the case of Saudi Arabia.

### 3.7. Bounds Test

To identify the existence of the covariance relationship (long-run relationships) in the (ARDL) model, the bounds test is used, and the significance of this test is recognized by its F-Statistic value (Nkoro and Uko 2016).

Table 7 indicates that there is no cointegration relationship, since the computed (F-statistic) value is less than the lower bound, I(0) of the critical values at the 5% significance level.

**Table 7.** Bounds test result.

| F-Bounds Test | | Null Hypothesis: No Co-Integration Relationship | | |
|---|---|---|---|---|
| Test Statistic | Value | Signif. | I(0) | I(1) |
| F-statistic | 0.04128 | 10% | 2.2 | 3.09 |
| k | 4 | 5% | 2.56 | 3.49 |
| | | 2.5% | 2.88 | 3.87 |
| | | 1% | 3.29 | 4.37 |

Thus, we accept the null hypothesis $H_0$, which states that there is no co-integration; that is, the absence of a long-run equilibrium relationship.

*3.8. ARDL Diagnostic Tests*

For diagnostic tests of residual distribution, autocorrelation, and identification problems—returning to Table 8, the results indicate that $H_0$ is accepted from the Jarque–Bera statistic, so the residuals of this model are normally distributed lines in Figure 3 (Gujarati 2009; Rufus Carter Hill 2011) and $H_0$ acceptance from the Breusch-Godfrey (BG) test for LM serial correlation (autocorrelations) states that there is no serial correlation (Greene 2018; Gujarati 2009). $H_0$ acceptance of the Ramsey RESET test (regression specification error test) was used to detect general functional form misspecification (Wooldridge 2015). However, regarding the heteroscedasticity (Gujarati 2009), it failed to reject $H_0$ from the Breusch–Pagan–Godfrey (BPG) test. Figure 4 shows the plots of the critical lines with 5% for CUSUM and CUSUMSQ to test the structural stability of the model's estimated parameters.

**Table 8.** ARDL diagnostic tests result.

|  | Test | Value | Probability |
|---|---|---|---|
| Residuals Distributed | Normality Test Jarque–Bera | 3.270557 | 0.173835 |
| Serial Correlation | LM Test/Breusch–Godfrey (BG) | 1.128449 | 0.309635 |
| Heteroskedasticity | Breusch–Pagan–Godfrey Test (BPG) | 18.19303 | 0.023662 |
| Stability | Ramsey RESET Test | 0.631393 | 0.407761 |
|  | CUSUM TEST | — | — |
|  | CUSUMSQ TEST | — | — |

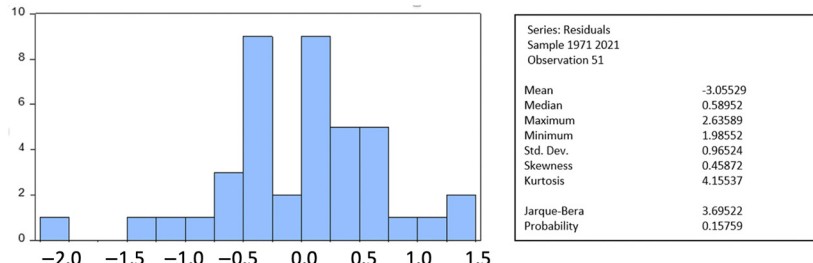

**Figure 3.** Histogram of residuals—normal distribution.

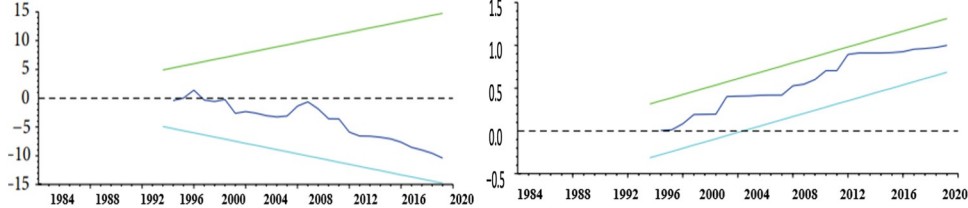

**Figure 4.** The plots of cumulative sum of recursive residuals and cumulative sum of squares of recursive residuals. The straight lines represent critical bounds at a 5% significance level.

## 4. Discussion and Conclusions

From the results of the estimation in the short run, the study found that whenever the gross domestic product (LGDP) increases by one unit, carbon dioxide emissions increase by about (6%), and we can explain this as a result of emitting economic activities; as for foreign direct investment (LFDI), the relationship is inverse with emissions. Whenever (LFDI) decreases by one unit, (LCO2) increases by about (0.09%), and this may be explained by increased restrictions of the green policy for the environment, while the variables (LMV) and (LTR) indicate no impact on carbon emissions in the short term. As for the results of long-run estimates, there is a direct effect of (LGDP) and manufacturing (LMV) on $CO_2$

emissions (LCO2). The greater that (LGDP) is by one unit, the greater that (LCO2) is by about (2.6%), and the greater that (LMV) is by one unit, the greater the (LCO2) is by (1.2%). As a result of this estimate, we reject the null hypothesis of the study that the source of growth in the Saudi economy, because of manufacturing, does not result in an increase in long-term environmental pollution. Additionally, the result of estimating the elasticity modulus of the GDP and referred to by the symbol (LGDP) in the short term is greater than in the long term, according to the values (6.977453) and (13.27487), respectively, and accordingly, it is clear that the phenomenon of the environmental Kuznets curve does not exist in Saudi Arabia, which is consistent with the study (Samargandi 2017), and from this point of view, we reject the null hypothesis of the study that the environmental Kuznets curve (EKC) applies to the state of the Saudi economy. From the result of the F-statistic of bounds test and estimated at a value of (3.031421), which is less than the minimum I(0) at a level of significance of 1%, there is no co-integration relationship between the variables in the long term, which confirms the absence of the Kuznets environmental curve (EKC) in Saudi Arabia.

As for the diagnostic tests in determining the efficiency of the model, the Residuals Diagnostics tests, which are the Serial Correlation LM test, the Normality test, and the Heteroskedasticity test (as in Table 8), the result of the Normality test indicates that the residual series follows the normal distribution, as shown in Figure 3, where the value of the probability of the Jarque–Bera statistic was estimated at 3.429695 and it is insignificant at the 5% level, and therefore we accept the null hypothesis by following the residual series for the normal distribution. Through the Serial Correlation LM test, the null hypothesis was accepted that the residual series does not suffer from the autocorrelation (serial correlation) problem, as the Breusch–Godfrey (BG) statistic was estimated at 1.183357, which is not significant at the 5% level. For the Heteroskedasticity test, the value of the Breusch–Pagan–Godfrey test (BPG) was estimated at 19.07826, which is significant, and led to us rejecting the null hypothesis in the presence of homogeneity of residual series variance. As for the stability tests of the structure of the model, and to ensure its stability and robustness, some tests must have been done. It was found through the Ramsey RESET test that the probability of the t- and F-statistics were estimated at a value of 0.4222, which is not significant at the 5% level. Therefore, we accept the null hypothesis that the model does not suffer from misspecification, and for the model and its parameters to be stable when repeating the sampling, the curve in Figure 4 for the CUSUM test and CUSUM of squares test must have been between the critical lines at a level of significance of 5%.

This study is an attempt to explore the impact of manufacturing on increasing emissions of carbon dioxide, and to verify the existence of the environmental Kuznets hypothesis in Saudi Arabia, but this study needs re-adjustment and greater dealing with the variables to be appropriate with the EKC model, and then it can be relied upon as a model that is characterized by econometric robustness and capable of estimating the relationship among the variables in the long-run and the short-run. We confirm that this study provided the steps followed to use the ARDL methodology using the Kuznets environmental curve model and the expected path to solve some problems of the model, and it does not claim that it addressed all the econometric problems, and the optimal interpretations of the results for the objectives of this study.

This is attributed to the increase in manufacturing activities and their diversity according to the goals of Saudi Arabia 2030. In addition, the model according to the selected variables may enable researchers interested in the field of environmental economics to provide more accurate results through the development of this study.

**Author Contributions:** Conceptualization N.A.; methodology, Y.O.; software, Y.O.; validation, I.E.; formal analysis, Y.O.; investigation, resources, N.A. data curation, N.A.; writing—original draft preparation, N.A.; writing—review and editing, Y.O. and I.E. All authors have read and agreed to the published version of the manuscript.

**Funding:** This research was funded by the Deanship of Scientific Research at KFU: GRANT 1984.

**Data Availability Statement:** This study analyzed publicly available datasets. These datasets can be accessed here: (World Development Indicators: https://databank.worldbank.org/source/world-development-indicators (accessed on 6 January 2023)).

**Conflicts of Interest:** The authors declare no conflict of interest.

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
