# Peer review of "Implications of Transition towards Manufacturing on the Environment: Saudi Arabia’s Vision 2030 Context"

_jrfm, doi:10.3390/jrfm16010044_

Round 1

Reviewer 1 Report

This manuscript studies the transition status of Saudi Arabia's Vision 2030 on manufacturing sector. The manufacturing and environment nexus is investigated to explore the Kuznets hypothesis using empirical regression time-series models. Overall, the topic of this manuscript is of interest, but some parts should be improved or clarified before it could be considered for publication. Here are the detailed comments:

1-The unique contributions of this manuscript are unclear to the reviewer, and I suppose our readers might also be confused to understand the innovation. The second paragraph of problem statement seems to mention about this aspect, but the statement reads weird, and the unique contributions are not properly identified.

2-In the correlation analysis, the authors are using correlation matrices here. A concern is on the preprocessing. Have you made lag-1 or other preprocessing on the time series data before calculating the correlation coefficients? If not, the high correlation might be related to the similar increasing trend which has limited insights in understanding the causality.

3-Why the Autoregressive Distributed-Lag (ARDL) is a suitable method for this regression? There are a large number of time-series regression models in the literature, so giving a proper reason for you model selection could be very important.

4-What is the relationship between the short-term and long-term regression models? It seems that they look similar, but the detailed relation is unclear to me. More explanations are needed in this aspect.

5-The current literature review is short and incomplete to capture the latest progress. A direct evidence is that the reference list is very short when compared to many other literature in this aspect. The authors are suggested to extend this part to well identify the current knowledge gap with more evidences.

Author Response

Response to the reviewer’s comments

Reviewer 2 Report

This study focusses on testing the EKC hypothesis using ARDL model on Saudi Arabia data. My concerns are as following:

1/This hypothesis has been widely tested in the literature, including the case of Saudi Arabia. The contribution stated in lines 67-69 is too general and does not show how this study is different from the reviewed studies on Saudi Arabia in lines 149-177.

2/For the interpretation of Table 6 (lines 281-305), could the authors recheck the magnitude of the sensitivity (e.g., 6%, 0.09%, 0.26, 2.6%, 1.2%) and explain where these magnitudes come from. For the case of GDP variable, its relationship with Co2 emission is a quadratic functional form while the author interpreted as a linear relationship, and hence the authors need to correct the interpretation for this variable.

3/ Several sentences do not make sense and there are several grammar and spelling errors (e.g., lines, 214, 230-233, 238, 245-246, 314, 344-345 ….). The authors need to carefully reread the study to correct them.

Author Response

Response to the reviewer’s comments
